# Assessing the Effectiveness of Supplemental Irrigation to Improve Soil Moisture in an Arid Ecosystem with an Emphasis on Climate Change: A Case Study from the State of Kuwait

**Ahmed Alqallaf** [1],*  , **Bader Al-Anzi** [2],*  **and Meshal Alabdullah** [3,4],*

1   Department of Environmental Science, Graduate Program, Kuwait University, Kuwait City 13060, Kuwait
2   Department of Environmental Technologies and Management, College of Life Sciences, Kuwait University, Kuwait City 13060, Kuwait
3   Department of Ecology and Conservation Biology, Texas A&M University, College Station, TX 77843, USA
4   Natural Environmental Systems and Technologies (NEST) Research Group, Ecolife Sciences Research and Consultation, Hawally 30004, Kuwait
*   Correspondence: ahmed.alqallaf89@gmail.com (A.A.); bader.alanzi@ku.edu.kw (B.A.-A.); m.abdullah_80@yahoo.com (M.A.); Tel.: +965-5023-8801 (A.A.)

**Abstract:** Arid ecosystems are extremely vulnerable to climate change, which is considered one of the serious global environmental issues that can cause critical challenges to the hydrological cycle in arid ecosystems. This work focused on assessing the effectiveness of supplemental irrigation to improve the actual soil moisture content in arid ecosystems and considering climate change impacts on soil moisture. The study was conducted at two fenced protected sites in Kuwait. The first site is naturally covered with *Rhanterietum epapposum*, whereas the other study site is a supplemented irrigated site, containing several revegetated native plants. The results showed that supplemental irrigation highly improved soil moisture (ΔSM) during the winter season by >50%. However, during the summer season, the rainfed and irrigated site showed low ΔSM due to the high temperature and high evapotranspiration (ET) rates. We also found that ΔSM would negatively get impacted by climate change. The climate change projection results showed that temperature would increase by 12%–23%, ET would increase by 17%–19%, and precipitation would decrease by 31%–46% by 2100. Such climate change impacts may also shift the current ecosystem from an arid to a hyper-arid ecosystem. Therefore, we concluded that irrigation is a practical option to support the ΔSM during the low-temperature months only (spring and winter) since the results did not show any progress during the summer season. It is also essential to consider the possibility of future shifting in ecosystems and plant communities in restoration and revegetation planning.

**Keywords:** arid ecosystem; climate change; supplemental irrigation; restoration; soil moisture; water budget

## 1. Introduction

Freshwater is considered the bloodstream of the biosphere as it is needed to drive critical ecosystem processes and functions that provide numerous essential ecosystem services, including supporting, provisioning, and regulating agricultural services [1]. However, population growth and anthropogenic activities are significantly affecting water security, including the quality and quantity of water resources. Water resources in arid ecosystems are dramatically influenced by human activities and climate, resulting in desertification, soil compaction, and the loss of ecosystem services, which,

in turn, decreases the water resources in these regions [2,3]. As a result of these anthropogenic activities, natural ecological systems are becoming more degradable, which is affecting the hydrological cycle [4–6]. Therefore, ecosystem restoration and management are now becoming key components for the long-term sustainability of natural resources, and this has been recognized globally, particularly in arid ecosystems [7,8].

Climate change is one of the serious global environmental issues that can cause critical challenges to the hydrological cycle in arid ecosystems [9]. This illustrates that there is an urgent need to understand the effects of such changes on the water budget in arid ecosystems [10]. Changes in the water budget may lead to drastic impacts on the desert ecosystems, including water storage, groundwater recharge, and the available water for native desert vegetation and agriculture irrigation. It was predicted that due to the lack of water for irrigated agriculture, 55% of the world population would live in countries incapable of self-sufficient food production by 2025 [11]. It was also illustrated that hydrological balance was highly influenced by climate and land use changes [12]. Therefore, supplementary irrigation is being used in some restoration and revegetation programs due to low precipitation (P), especially with climate change impacts that are affecting the water storage, evaporation, and evapotranspiration (ET) rates [13]. However, irrigation programs need to be appropriately planed due to the scarce water resources in these regions; as well, over-watering may support vegetation survival initially, but increase soil salinity and negatively impact plant diversity over the long term [14].

In order to provide a sustainable irrigation plan, it is necessary to develop a proper understanding of future uncertainties by determining the sufficient time for irrigation and the water requirements for native desert plants. It was recommended by He [15] that supplementary irrigation patterns should be adjusted during the high-temperatures season (summer season) to provide sufficient moisture for healthy plant growth. However, it may not always be true, especially when dealing with arid ecosystems, which are considered as having high potential ET rates and low precipitation [16]. Previous studies showed that nearly all rainfall in arid ecosystems evaporates back to the atmosphere [17,18], and only 3%–50% of the annual rainfall remains in the soil [19,20]. Therefore, understanding the relationship between precipitation and ET and their spatiotemporal distribution is essential for the sustainable management of scarce water resources. Our understanding of irrigation's benefits in improving the soil moisture content in arid landscapes is still lacking. Thus, this work aims to assess the effectiveness of supplemental irrigation to improve the actual soil moisture content in arid ecosystems. Specific objectives are to (1) apply a water budget model for a rainfed protected area and a supplemental irrigated protected area to understand the seasonal behavior of the soil moisture content at these sites, and (2) predict the impacts of future climate changes on the soil moisture content. The outcome of this work would be essential for landscape management and restoration and revegetation planning, especially when considering future uncertainties such as climate change, which may negatively influence the hydrological processes.

## 2. Materials and Methods

### 2.1. Study Area

This work was conducted in two protected sites located at Al Abdali farms, with a total area of 6 hectares for each site. Al Abdali is situated in the northern part of the State of Kuwait and is considered an agricultural area (Figure 1). It is considered an arid ecosystem with very hot summers and cold winters with limited rainfall events. The first study site is fenced and naturally covered with native desert plants and dominated by *Rhanterietum epapposum* (perennial shrub), and the soil type is sandy clay loam, whereas the second study site is a supplemental irrigated fenced site and also contains native revegetated plants using drip irrigation, and has sand and loamy sand soil. According to Abdullah M M et al. [21], the total vegetation cover in the unirrigated site was 8.5% in October 2018, which was divided into 1.9% annual plants and dominated by 6.6% *Rhanterietum epapposum*. However, after the rainy event in November 2018, the vegetation distribution significantly increased in December

2018 and January 2019 to reach 24% and 57%, respectively. The dominant soil type in the study areas was the loose sandy soils. The surface soil is generally pale brown while the subsoil, which consisted of a limestone mixture, was very pale brown.

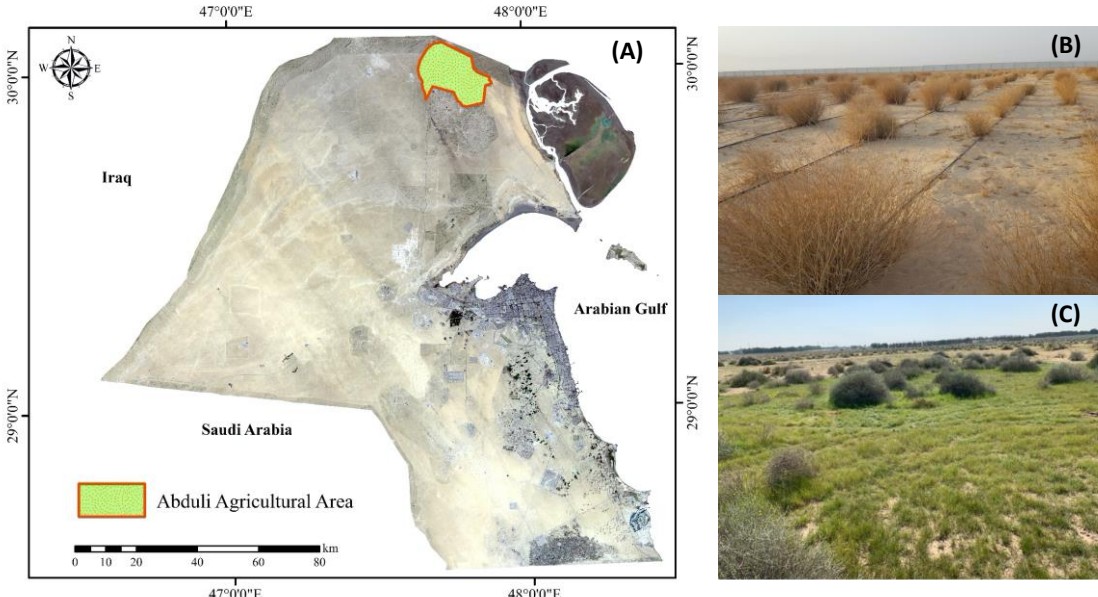

**Figure 1.** Study area. (**A**) Map of State of Kuwait and location of the Al Abdali farms where the study is conducted. (**B**) A photo of the revegetated site. (**C**) A photo of the natural vegetation site.

*2.2. Implementing the Water Budget Model*

We examined the monthly rainfall for five rainy years for the period from 2013 to 2017. According to rainfall variation and fluctuation in arid regions, we classified the rainy year into wet, average, and dry years following the yearly rainfall averages for the ten-year period from 2008 to 2017. Then, a conceptual water budget model for the five examined years was applied at both study sites to simulate and quantify monthly soil moisture content in order to evaluate the effectiveness of using irrigation techniques to increase moisture content to support the growth of native desert plants. The water budget is based on the mass and energy conservation principle and can be defined as the change in water quantity for a specific control volume over time [22].

$$\text{Change in Soil Moisture} = \text{Inflow} - \text{Outflow} \tag{1}$$

In arid ecosystems, changes in soil moisture (ΔSM) in the root zone are highly influenced by precipitation, evapotranspiration (ET), and runoff (Q). The hydroclimatic model was utilized to estimate the water budget of the study sites (Figure 2), which was calculated using the following equation:

$$\Delta SM = P - ET - Q \tag{2}$$

where ΔSM is in mm/day. However, for the irrigated fenced site, the following equation was utilized:

$$\Delta SM_{ir} = P + IR - ET - Q \tag{3}$$

where $\Delta SM_{ir}$ is the change in soil moisture in the irrigated area (mm/day) and *IR* is the amount of water used for irrigation (mm/day).

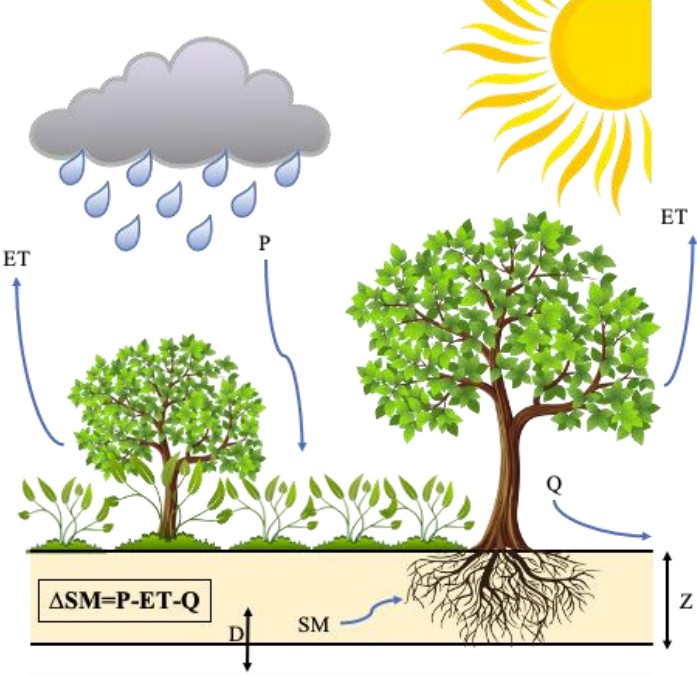

**Figure 2.** Conceptual model of the water budget parameters. Change in soil moisture (ΔSM), precipitation (P), evapotranspiration (ET), run-off (Q), drainage from the root zone (D) and unsaturated zone (0–30 cm) (Z).

Finally, the performance of the model was validated by considering the correlation between the model-predicted data and the soil moisture collected from the field for the year 2017 since it was the only year with available actual soil data. Regressions analysis using the JMP™ software [23] was performed to obtain the coefficient of determination ($r^2$) and probability value (*p*-value).

### 2.3. Data Collection and Preparation

#### 2.3.1. Meteorological Data

The meteorological data, including daily P (mm), temperature (T) (°C), wind speed (m/s), evaporation, and relative humidity (%) were obtained from the Directorate General of Civil Aviation (DGCA), Kuwait Airport Station and Al Abdali Station. The data were collected for 60 months from 2013 to 2017, which was used as the input to compute the runoff, ET, and water budget model.

#### 2.3.2. Soil Data Collection

A total of 36 soil samples were collected in October and February 2017 (before and after the rainfall season). Three plots (5 × 5 cm) were selected randomly as replicates at each site, and six samples were collected at each plot from the surface and 30 cm depth to measure grain size and soil moisture in the field. Soil moisture content was used to validate the model by comparing the model's results with the actual soil moisture data. However, the grain size was used as an input in the runoff model, which is discussed in detail in the runoff section. All soil samples were analyzed by the National Unit for Environmental Research and Services (NUERS) at Kuwait University. The actual ΔSM values were measured following ASTM D4643-17, the "Standard Test Method for Determination of Water Content of Soil and Rock by Microwave Oven Heating" [24]. However, the grain sizes were analyzed according to ASTM D422-63—the "Standard Test Method for Particle-Size Analysis of Soils" [25].

### 2.3.3. Supplemental Irrigation

The irrigated water data were collected from the Public Authority of Agricultural and Fisheries (PAAF). Drip water irrigation is used at the irrigated site, which contains revegetated dominant native vegetation, including *Rhanterietum epapposum*. The primary type of water used in the site is groundwater and desalinated water. The consumed irrigated water differs between summer, spring, and winter seasons. A total amount of 3.8 L/m$^2$/day is implemented twice a week during the spring and winter seasons, where the temperature is low. However, irrigation takes place twice a day (early morning and afternoon) on a daily basis during the summer seasons due to the high temperature (>45 °C).

### 2.3.4. Implementing Model Parameters

Several parameters were considered in this work to estimate the water budget in the study sites. Data for each parameter, including water runoff (Q), ET, amount of water used for irrigation, and water drainage from the root zone (D) and beyond the root zone (Z) were collected to develop the water budget model. However, drainage in the study area is negligible because the characteristics of the dryland vegetation take up the water before it can move beyond the root zone by the deep roots of plants [26,27].

- Surface water runoff (Q)

Surface water runoff was estimated using the Soil Conservation Service Curve Number Equation (SCS-CNE) model developed in the mid-1950s [28,29]. The model mainly depends on estimating the relationship between the initial abstraction and runoff as a function of soil type and land use. The precipitation–runoff relationship was calculated using the following equation [30]:

$$Q = \frac{(P - I_a)^2}{(P - I_a) + S} \tag{4}$$

where *Q*—run-off (mm), *P*—precipitation (mm), *S*—potential maximum retention after runoff begins (mm), and I$_a$—initial abstraction (mm). Assuming that the initial abstraction is equal to 20% of the potential maximum retention (*I$_a$* = 0.2*S*) as suggested by Al-Dousari et al. [31]. The above equation can be simplified to:

$$Q = \frac{(P - 0.2S)^2}{(P + 0.8S)} \tag{5}$$

where, as *Q* depends on the *P* intensity, as stated by Mockus [32]:

$$Q = \begin{cases} 0 \; for \; P \leq 0.2S \\ \frac{(P-0.2S)^2}{(P+0.8S)} \; for \; P > 0.2S \end{cases} \tag{6}$$

where *S* is related to the soil and cover conditions through the curve number (*CN*), which is an empirical parameter used in hydrology for predicting direct runoff or infiltration from rainfall excess [33]. S is derived from *CN* by:

$$S = \frac{1000}{CN} - 10 \times 25.4 \tag{7}$$

The *CN* is based on hydrologic soil groups (HSGs), land use, and hydrologic condition [34]. The actual soil grain data were used as an input to estimate the surface runoff model.

- Evapotranspiration

The pan evaporation (Epan) method was used in this study to estimate potential ET. Potential ET represents the atmospheric water demand using the pan evaporation (Epan) method. This method has shown consistent results and was successfully used to determine ET according to the evaporation

loss [35]. The Epan value depends on the relative humidity, temperature, and wind speed [36]. The ET equation is expressed as:

$$ET = k_p \times E_{pan} \tag{8}$$

where *ET*—evapotranspiration (mm/day), $k_p$—pan coefficient, and *Epan*—pan evaporation (mm/day).

### 2.3.5. Determining the Main Variables Influencing the Water Budget Model

The model results were then used to determine the significant metrological parameters influencing the soil moisture content using statistical tests within JMP™ statistical software. Simple linear regression was performed on each parameter individually to identify the correlation between the metrological parameters and the soil moisture content. This was then followed by the multivariate stepwise regression analysis to determine the best fit model that combined multiple factors. The multivariate stepwise regression analysis was performed twice in this work. The first run combined all factors including the precipitation, temperature, surface runoff, and ET. However, precipitation was excluded from the second run to avoid any multicollinearity, which may impact the results since all variables mainly depend on precipitation.

### 2.3.6. Climate Change Prediction

The climate change projection was implemented in this work to determine future changes in precipitation, temperature, and evapotranspiration. The statistical downscaling method was used in this work to project future precipitation and temperature. This method was implemented and proven by several studies, which showed similar results to the dynamical downscaling approach [37,38]. Future precipitation and temperature were created based on Kuwait's Second National Communication (KSNC) using global circulation models (GCMs). The GCMs' output under the historical scenario for the period 1986–2005 was obtained, and two Representative Concentration Pathways (RCPs) scenarios (RCP4.5 and RCP8.5) were implemented for the period 2021–2100 [39]. RCP4.5 is related to the low-medium stabilization scenarios [40], and RCP8.5 is related to very high baseline greenhouse emission scenarios [41]. Future temperature (°C) and precipitation of 2021–2100 for RCP4.5 and RCP 8.5 scenarios were computed based on the baseline years 1986–2005. The 2021–2100 period was divided into four quarters as 2021–2040, 2041–2060, 2061–2080, and 2081–2100. Afterwards, the future climate data for each GCM were downscaled by implementing a time series with random years, which was selected according to the baseline data. Random year was selected for each quarter from 2021 through 2100, following the baseline years (1986–2005). Finally, the anomalies of the temperature (°C) and precipitation were added to the baseline data using the following equation:

$$X_{F\,i} = X_{B\,i} + \Delta X_i \ (9) \tag{9}$$

where

$X_{F\,i}$ is the future value of the temperature or precipitation on year *i*.
$X_{B\,i}$ is the baseline value of the temperature or precipitation on year *i*.
$\Delta X_i$ is the value of anomaly of the temperature or precipitation on year *i*.

However, projected ET was computed according to the Thornthwaite model [42] by using the historical and forecasted metrological data obtained from GCMs. This method is known as the delta change method of downscaling climate change data for a specific location [43]. The following equation was implemented to project the ET:

$$PET = 16\left[\frac{10tn}{J}\right]^a \tag{10}$$

where:

$$J = \sum_{1}^{12} j \tag{11}$$

$$j = \left[\frac{tn}{5}\right]^{1.514} \tag{12}$$

$$a = 0.016J + 0.5 \tag{13}$$

where *PET*—potential evapotranspiration, *J*—heat index, *j*—coefficient monthly temperature (°C), a—constant, and tn—average monthly temperature (°C).

Finally, the results of the climate change projection of the precipitation, temperature, and ET were used as inputs in the water budget model to estimate surface runoff and soil moisture content to determine the climate change impact on the soil moisture content.

### 2.3.7. Aridity Index

At this stage of the work, the aridity index (*AI*) was utilized to classify the current degree of aridity to determine possible future shifting in the ecosystem as a result of climate change. *AI* is an indicator that shows the degree of dryness based on climatic classifications in a specific location. In arid regions, the *AI* ranges between 3% and 20%, while areas with an *AI* less than 3% are classified as hyper-arid regions. The *AI* formula depends primarily on the annual *P* (mm/year) and annual *ET* (mm/year) [44].

$$AI = \frac{P}{ET_o} \tag{14}$$

## 3. Results

### 3.1. Rainy Seasons Classification

According to the rainfall analyses, the rainfall data during the study period 2013–2017 were classified into wet, average, and dry years relative to the historical average rainfall data from the years 2008–2017. The total rainfall of 2015 (89.6 mm) and 2017 (87.2 mm) were above the rainfall average of 68.0 mm and classified as wet years, 2014 (53.2 mm) and 2016 (50.0 mm) are considered as average rainy years, and 2013 (42.4 mm) was below the average rainfall and classified as a dry year (Figure 3).

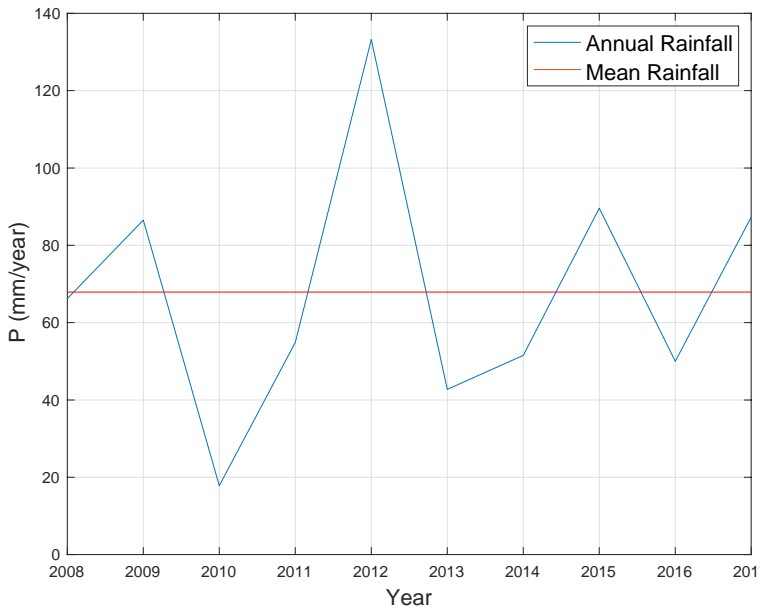

**Figure 3.** Historical and average data of the rainfall in the Al Abdali farms from 2008 to 2017.

### 3.2. Seasonal Behavior of Water Budget Model

The results showed a clear correlation between the modeled and actual soil moisture collected from the field in 2017 ($R^2$ = 0.9) (Figure 4). This illustrates that the model provides reasonable results to measure and evaluate the soil water storage in arid landscapes.

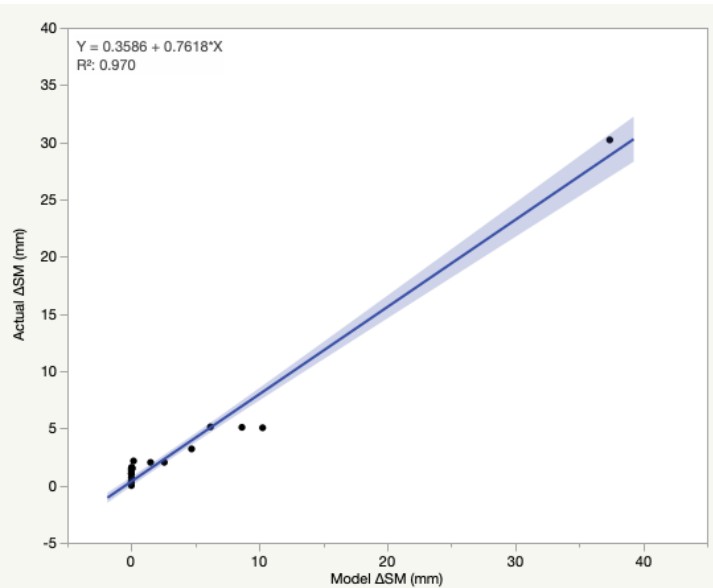

**Figure 4.** Relationship between modeled soil moisture (ΔSM) values and predicted soil moisture for the year 2017.

It was illustrated from the results that the ΔSM variations were mainly influenced by rainfall fluctuation according to the examined years. The ET rate differed according to the yearly seasons with an average of 40 mm during spring and winter seasons (low-temperature seasons) and increased significantly to >100 mm during the hot, dry months (May–September) (Figure 5c,f). However, runoff was mainly influenced by the wet season, showing a small value of 4.92 and 9.66 mm in the high precipitation years of 2015 and 2017, respectively (Figure 5e).

### 3.3. Rainfed vs. Irrigated Site

The results showed that ΔSM differed between the rainfed and irrigated sites only during the rainy period (January–April and October–November), including spring and winter seasons. However, during the summer season, hot and dry months with shallow rainfall events (May–September), the ΔSM for both sites showed zero values (Figure 5a,b). It was also found that the seasonal ΔSM and ΔSMir differed between the classified years, including the wet, average, and dry years.

#### 3.3.1. Wet Years (2015 and 2017)

The results showed that during the rainy months, the ΔSM surplus was high at the rainfed site, estimated to be 41.5 mm in 2015 and 46.41 mm in 2017. The results showed zero ΔSM during the dry months (May–September), with a yearly average of 5.51 mm/year. However, at the irrigated site, the $\Delta SM_{ir}$ was 16% more at the rainfed site, where the ΔSM reached 72.22 mm in 2015 and 60.87 mm in 2017 during the rainy months, but during the dry months, the $\Delta SM_{ir}$ showed little positive change (1.00 mm) in 2015 and 1.72 mm in 2017.

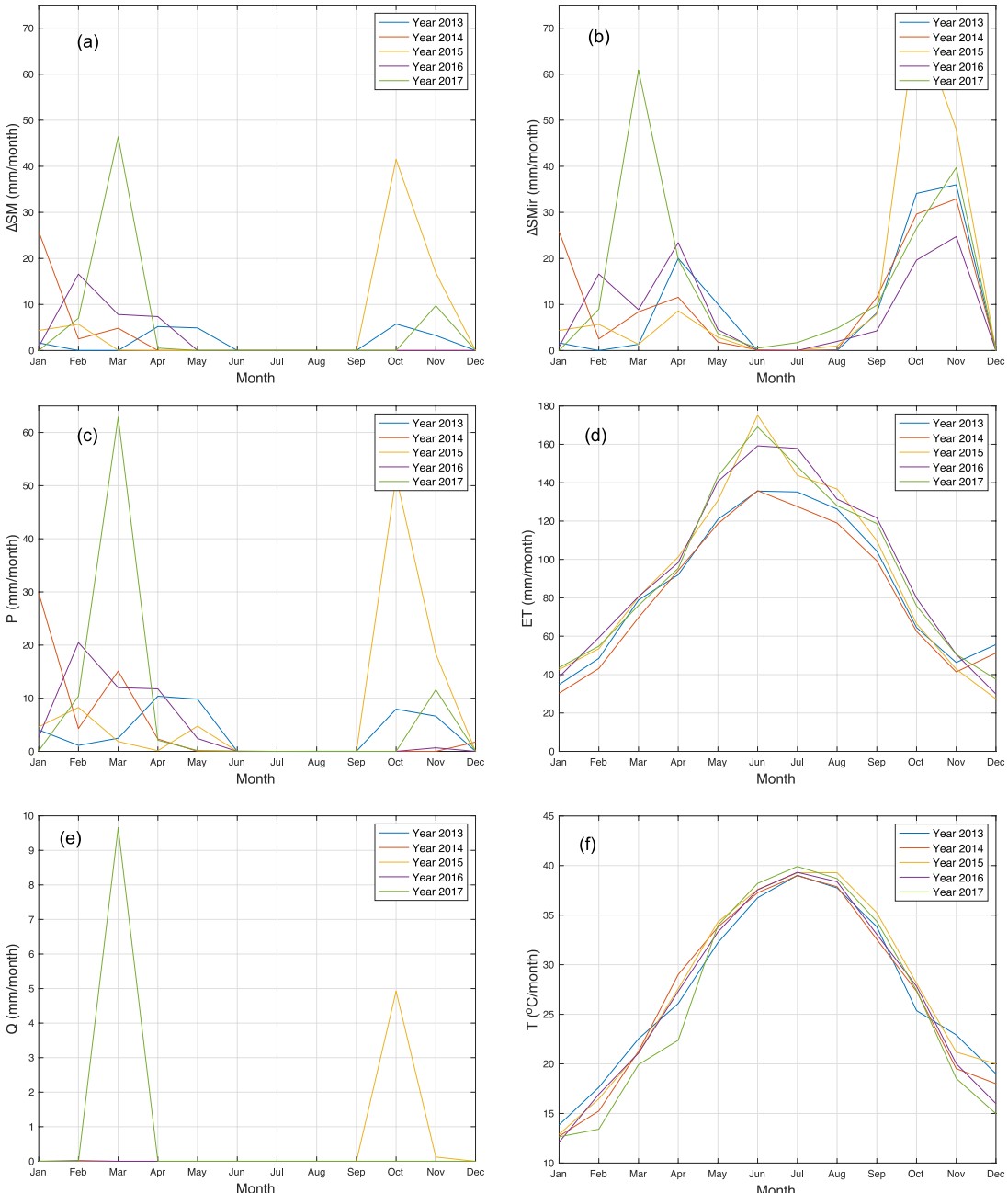

**Figure 5.** Model results for the period of 2013–2017 in Al Abdali farms: (**a**) monthly change in soil moisture (ΔSM) (mm/month) for the rainfed area, (**b**) monthly change in soil moisture (ΔSM) (mm/month) for the supplemental irrigated area, (**c**) monthly precipitation (P) (mm/month), (**d**) monthly evapotranspiration rate (ET) (mm/month), (**e**) monthly surface run-off (Q) (mm/month), and (**f**) monthly average temperature (T) (°C/month).

### 3.3.2. Average Rainfall Years (2014 and 2016)

During the average rainfall years (2014 and 2016), the ΔSM was lower during the rainy months at the rainfed site compared with the wet years, which reached 25.84 mm in 2014 and 16.60 mm in 2016 with a yearly average of 2.77 and 2.72 mm/year, respectively. However, during the dry months, the ΔSM was similar to the wet years, where no significant amount was recorded. While at the irrigated site, ΔSM$_{ir}$ increased by 26% (32.91 mm in 2014 and 24.74 mm in 2016) with an average of 10.38 mm in 2014 and 8.74 mm in 2016.

### 3.3.3. Dry Year (2013)

The year 2013 was considered a dry year as the average amount of rainfall was 3.53 mm. The results showed that the average ΔSM at the rainfed area during the rainy months (10.76 mm) was 1.73 mm, while, during the dry months, the ΔSM was 0 mm. However, the $\Delta SM_{ir}$ reached 36.21 mm at the irrigated site in October and November with a yearly average of 9.28 mm, which illustrates the ability of irrigation to support soil moisture content during the spring and winter seasons.

### 3.4. Factors Influencing the Water Storage Model

The simple linear regression analysis showed that the ΔSM was significantly correlated with precipitation and surface runoff (Figure 6). The simple linear regression analysis showed that precipitation is considered the most influential factor ($r^2 = 0.976$), followed by surface runoff ($r^2 = 0.669$). However, temperature and ET showed a lower correlation with the ΔSM ($r^2 = 0.342$ for temperature and $r^2 = 0.253$ for the ET).

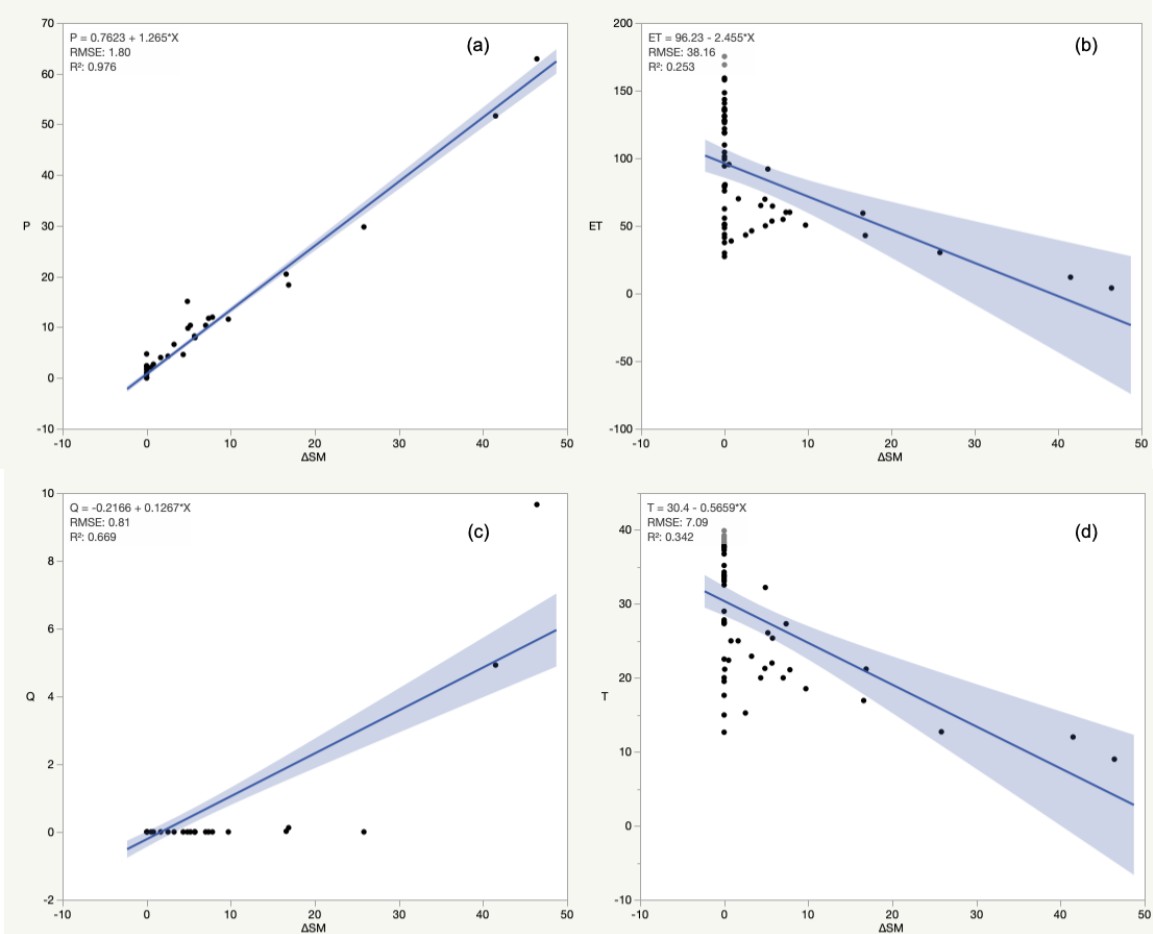

**Figure 6.** Simple linear regression for each parameter individually to identify the correlation between the metrological parameters and the soil moisture content. (**a**) Correlation between soil moisture ΔSM and precipitation, (**b**) correlation between soil moisture ΔSM and ET, (**c**) correlation between soil moisture ΔSM and Runoff (Q), and (**d**) correlation between soil moisture ΔSM and Temperature.

However, the forward stepwise regression analysis, including all factors, showed that the best fit model included only the precipitation $r^2 = 0.976$ and *p*-value < 0.0001. Such results make sense since all other factors mainly depend on the precipitation. Therefore, when precipitation was excluded from the analysis, the results showed that the best fit model included surface runoff and temperature with an $r^2 = 0.089$ and 0.758, and *p*-value < 0.0001 (Table 1).

**Table 1.** Multivariate correlation: SM is the dependent variable while P, ET, and Q are the independent variables.

| | Independent Variable | Multivariate, Stepwise | |
| --- | --- | --- | --- |
| **With Precipitation** | **Parameter** | **R2 Added** | ***p*-Value** |
| | P | 0.976 | 0.0001 |
| | Q | 0.0002 | 0.5788 |
| | T | 0.0003 | 0.5976 |
| | ET | 0.0003 | 0.9528 |
| **Without Precipitation** | Q | 0.089 | 0.0001 |
| | T | 0.758 | 0.0000 |
| | ET | 0.000 | 0.7260 |

### 3.5. Climate Change Projection for the Period 2021–2100

The results of the future projection showed that the water budget and ΔSM would be significantly impacted by the year 2100 due to the changes in temperature and precipitation. In the first quarter (years 2021–2040), the mean temperature will increase uniformly for both scenarios (RCP4.5 and RCP8.5) by 0.45 °C, reaching an average of 28.4 °C. Consequently, the mean temperature for the next quarter (2041–2060) will increase by 0.5 and 1.7 °C for RCP4.5 and RCP8.5, reaching 28.5 and 30.0 °C for RCP4.5 and RCP8.5, respectively. Likewise, the third quarter (2061–2080) showed a continuity increase in the mean temperature for both scenarios by 0.5 and 1.3 °C for RCP4.5 and RCP8.5, respectively. Then, it will reach 29.0 and 31.5 °C for both RCP4.5 and RCP8.5 at the end of the third quarter. The mean temperature for the last quarter (2081–2100) showed a continued increase, reaching 30.25 and 33.12 °C for RCP4.5 and RCP8.5, respectively. The total increase in the temperature from 2021 to 2100 was estimated to be 12% (+3.4 °C) and 23% (+6.5 °C) for RCP4.5 and RCP8.5, respectively (Figure 7a).

The increase in temperature was associated with precipitation variation. In the first quarter (years 2021–2040), the mean precipitation may increase to 95 and 108 mm for RCP4.5 and RCP8.5. However, in the second quarter (years 2041–2060), the mean precipitation will drop below the average annual precipitation (94.4 mm) to reach 72.5 and 68.7 mm for RCP4.5 and RCP8.5, respectively. Lastly, in the third and fourth quarters (years 2061–2100), the mean precipitation would continue dropping to 65 and 58 mm for RCP4.5 and RCP8.5. During the period 2021–2100, the projected mean precipitation will drop to reach the value of 20 mm for both the RCPs (Figure 7b).

Such changes in temperature and precipitation would significantly impact the evaporation and ET rates, and the surface runoff, leading to a dramatic decrease in the amount of ΔSM. The model showed that the ET rate would increase uniformly by 8% and 10% to reach the values of 1191 and 1236 mm/year in the year 2100 for RCP4.5 and RCP8.5, respectively (Figure 7c). It was also found that the results that the Q would increase remarkably in the first quarter of the year 2020–2021 (Figure 7d). The mean surface runoff will decrease in the first, second, third, and fourth quarters by 2.53, 1.54, 2.64, and 1.31 mm/year for RPC4.5, and 2.37, 1.03, 1.52, and 0.76 mm/year for RCP8.5. At the end of the century, the surface runoff would drop by 48% and 68% for RCP4.5 and RCP8.5.

Such significant changes in the water budget components would significantly influence the ΔSM (Figure 7e). The results showed that in the first quarter (years 2020–2040), the ΔSM would be fluctuating for both scenarios between −1104 and −877 mm/year. In the year 2041 and later, the ΔSM would start to drop down, reaching −1217 and −1266 mm/year at the end of the century for RCP4.5 and RCP8.5, respectively. The total ΔSM for RCP4.5 and RCP8.5 was estimated to be −12% and −16% in the year 2100. These changes may significantly shift the current ecosystem from an arid to a hyper-arid ecosystem by the year 2100, leading to more drought and desertification (Figure 7f). According to the climate change scenario results, the AI may increase from 4%–16%, which is attributed to the increase in precipitation. While, during the second and third quarters (years 2041–2080), the AI would Decrease from 9%–3%, which will be at the lower bound of the hyper-arid region. During the fourth quarter

(years 2081–2100), the AI for the RCP8.5 scenario was estimated to be 2% in 2088, which is at the hyper-arid region category, while RCP4.5 will stay between 3%–5% at the arid region category.

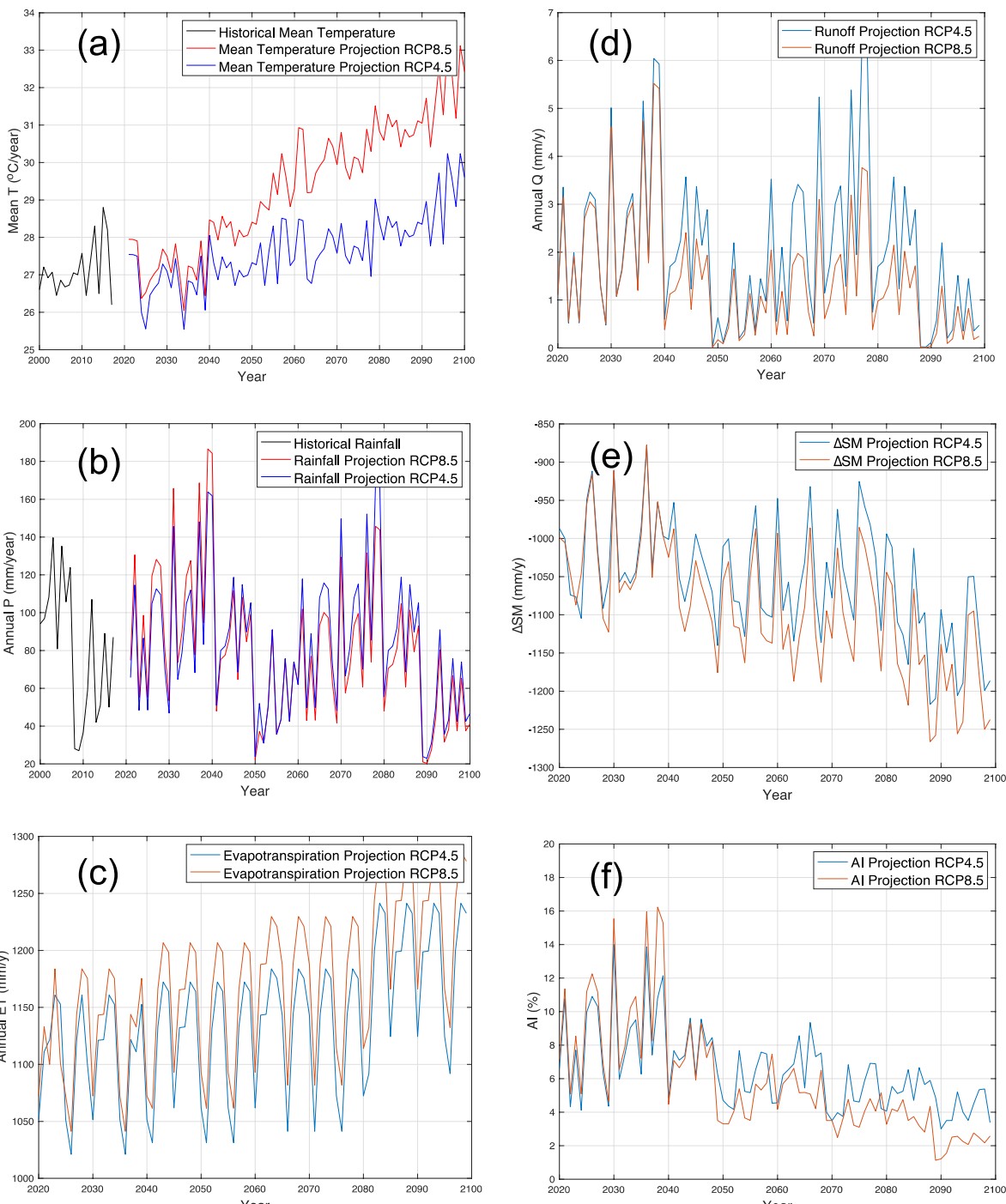

**Figure 7.** Climate change prediction for the period of 2020–2100 according to RCP4.5 and RCP8.5: (**a**) mean historical and predicted temperature (T) in (°C/year), (**b**) historical and predicted precipitation (P) (mm/year), (**c**) predicted evapotranspiration (ET) (mm/year), (**d**) predicted run-off (Q) (mm/year), (**e**) predicted change in soil storage (ΔSM) (mm/year), and (**f**) predicted aridity index (AI) (%).

## 4. Discussions

### 4.1. Seasonal Behavior of the Water Budget Model

This work presents fundamental findings to support natural ecosystems and restoration and revegetation planning as changes in the water budget will negatively influence arid ecosystems. It was demonstrated that ΔSM differed between the dry and wet seasons, as 90% of the ΔSM surplus (positive value) occurred during extreme rainfall events (January, March, and November). The ΔSM surplus generates surface runoff and/or groundwater recharge, which primarily occurs during the winter when the ET rate is lower than the precipitation. However, ΔSM deficits (negative values or zero values) indicate that the water outflow is more massive than the water inflow due to the severe water loss associated with the high ET rates, especially during the dry months [45].

Such a significant decrease in ΔSM during the summer season is highly related to the substantial increase in ET due to the low rainfall events and increase in temperature, which reaches 50 °C during the summer season. It was found that precipitation plays a vital role in enhancing the hydrological cycle, while high temperatures deplete the ΔSM through the ET process [46]. Therefore, water ET is considered the predominant component of the water budget model (ET > P) [47]. It was illustrated from the results that the total loss of precipitation due to ET was 94% and that ET was much higher than precipitation during the dry seasons (April–September). In contrast, during the hot seasons, monthly ET was greater than 100 mm, representing 71.6% of the total annual ET, where most of the ΔSM deficit occurred. However, monthly ET dramatically decreases to 40 mm during December and January, where temperatures decrease, providing more water storage for plant uptake. This illustrates that ET captures most of the inflow water infiltrated into the soil and is considered a major reason for ΔSM deficits during the summer season [47].

### 4.2. The Effect of Climate Change on the Water Budget Model

It was illustrated from the work that uncertainties such as climate change, considered a serious issue which will significantly influence the ΔSM in arid ecosystems, are leading to more drought seasons. The RCP scenarios showed significant variation in the projected temperature at the end of the century, expecting to cause a more substantial decrease in water budget components (Figure 8). The total change in mean temperature for 2021–2100 was 7%–15% higher for the RCP4.5 and RCP8.5 scenarios. A higher temperature would increase the ET rate and expand the summer seasons, which indicates that the atmosphere may require more water to meet its evaporative demand. The total ET was estimated to increase by 17%–19% for RCP4.5 and RCP8.5 in 2100, and the precipitation rate will decrease to 31%–46% by the end of the century.

Such future changes will lead to a higher vulnerability to drought; the biodiversity and native species will be at high risk, as well [48]. These changes will also impact the AI ratio; it seems that several areas may shift from arid to hyper-arid ecosystems. It was illustrated that in 2088, the projected AI would drop to less than 3%, that is, to the hyper-arid category, due to the increase of ET values and the decrease of precipitation quantity (ET > P). Such changes in the climate are likely to negatively affect the rate of water recharge, vegetation cover, and soil infiltration rate, which indicates possible aggravation of the water stress in the future. Such effects will impact the ecosystem balance and water availability in aquifers and cause a massive desertification disaster [49].

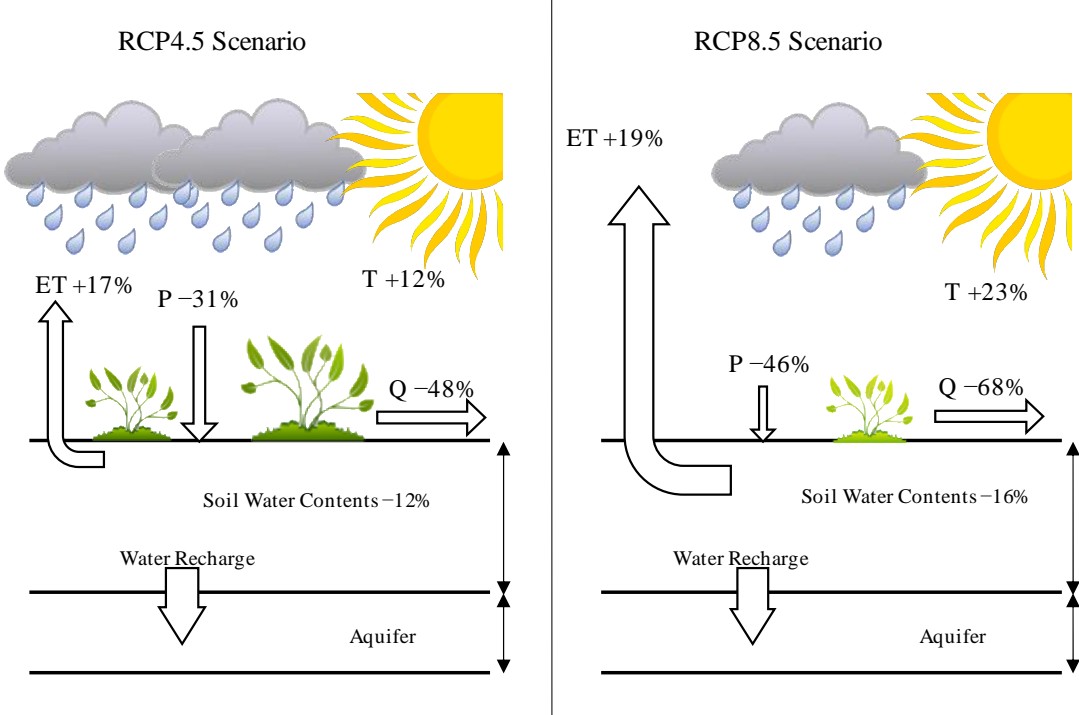

**Figure 8.** Change in water budget components due to climate change scenarios at the end of the century, the year 2100.

### 4.3. Can Irrigation Support the Soil Water Content in Arid Lands?

More attention has been given to the restoration of desert ecosystems in recent years due to the increase in desertification and climate change [46]. Our results showed that climate change might lead to a shortage in ΔSM in arid lands, which is a potential threat to the survival of native plants as it constrains the growth of native vegetation [50,51]. Therefore, irrigating native desert plants has been discussed as an option in restoration and revegetation planning to increase ΔSM and accelerate plant growth. According to Deng et al. [52], less than 600 mm per year of annual precipitation may impact the ΔSM, indicating that planned vegetation restoration is a poor choice in places where the annual rainfall is less than the potential ET. However, our results showed that irrigation is not always a practical option as it depends on the condition of the season. It was demonstrated from this work that irrigation is a valuable option only during low-temperature seasons, spring and winter, as ΔSM significantly increased by 50%, and the ΔSM deficits in the irrigated site were less than the rainfed site by 70%. However, irrigation did not support the ΔSM during the summer season due to the increase in ET rates. Still, it is critical to keep in mind that this option should be considered carefully to avoid any future complications as intensive watering may not be sustainable in the long term, leading to salinization of the soil [53,54]. This is important since increasing water availability through irrigation may provide a temporary success that is quickly followed by poor plant adaptation and eventual failure [55].

Future increases in temperature and ET rates as a result of climate change could also increase the summer season period and decrease the period of the vegetation growing season. We believe that summer irrigation could help during the early months of the summer season, where the temperature is <35 °C. However, it is unlikely to be useful during the mid-summer season, where the temperature reaches >45 °C. Therefore, it is crucial to adjust the irrigation pattern according to the water budget components, including temperatures, precipitation, ET, and surface runoff. It is also necessary to reconsider the types of plants in restoration and revegetation planning, as each vegetation type requires different adjustments [56]. We believe that further research work is needed to develop a better understanding of the climate-vegetation interaction.

*4.4. Research Limitation and Future Enhancement*

The model used in our experiment was a numerical model that primarily depended on the input and output of the hydrological parameters measured on a small scale. A large-scale assessment requires more parameters to investigate, including elevation, seepage, initial groundwater storage, and vegetation type. We also highly recommend that future research needs to focus on estimating the actual ET rate for native desert plants for a better water budget assessment. This is considered a significant challenge, as most studies focused on modeling the ET rate for agriculture areas. The mechanisms and dynamics of native arid plants need to be given more attention since they are highly vulnerable to climate change uncertainties. Supplementary irrigation has also been used in some restoration and revegetation programs in arid ecosystems. However, we still believe that more studies need to focus on the long-term impacts to determine future negative complications. Therefore, there is a critical need to develop more accurate models for natural arid ecosystems to build a better understanding of the water requirements for native desert plants.

**Author Contributions:** All authors conceived and designed the research; Supervision; B.A. and M.A.; A.A. performed the experiments and analyzed the data; A.A. and M.A. collected the data; B.A., A.A. and M.A. wrote and edited the manuscript.

**Funding:** This research received no external funding.

**Acknowledgments:** The authors would like to acknowledge all sources of the data used in this paper: the Directorate General of Civil Aviation (DGCA) for providing all required metrological data and the National Unit for Environmental Research and Services (NUERS) at Kuwait University for soil samples analysis. Finally, we acknowledge the Public Authority of Agricultural and Fisheries (PAAF).

**Conflicts of Interest:** The authors declare no conflict of interest.

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
