# Peer review of "Assessing the Effectiveness of Supplemental Irrigation to Improve Soil Moisture in an Arid Ecosystem with an Emphasis on Climate Change: A Case Study from the State of Kuwait"

_sustainability, doi:10.3390/su12219104_

Round 1

Reviewer 1 Report

This study investigated the influences of irrigation on soil moisture status in the arid ecosystem. In general, this manuscript was well organized and well written. However, there are several major concerns that need to be addressed before this manuscript can be moved forward.

(1) In the introduction, the previous studies on the impacts of field water management on soil moisture should have been thoroughly reviewed and identified the knowledge gap to highlight the novelty of this research. For example, L64-65: "it may not always be true", other's work should be cited and discussed to support your statement.

(2) Significance test should be conducted to make the comparison results more solid.

(3) It seems that the two research sites have different landcover. Since soil moisture is also affected by plants, this factor should be properly addressed.

(4) Only one year (2017) measure data is available, the results can be questioned. This part should be explained.

(5) Both model-predicted and measured soil moisture data were used in this study, in the results and discussion, which data was referer to should be clearly stated. For example, in the regression analysis, measured or predicted soil moisture data was used? L195: it seems that the modeled results were used in the regression analysis. If this was the case, the results can be biased.

Author Response

Reviewer 1:

This study investigated the influences of irrigation on soil moisture status in the arid ecosystem. In general, this manuscript was well organized and well written. However, there are several major concerns that need to be addressed before this manuscript can be moved forward.

(1) In the introduction, the previous studies on the impacts of field water management on soil moisture should have been thoroughly reviewed and identified the knowledge gap to highlight the novelty of this research. For example, L64-65: "it may not always be true", other's work should be cited and discussed to support your statement.

Response: revised as suggested. see line 66 and 67.

(2) Significance test should be conducted to make the comparison results more solid.

Response: It is unclear what you mean by significant test. I believe you might mean ANOVA test to determine whether the differences were significant. If this what the reviewer suggests, such tests are unapplicable since the ANOVA test compares the means. So, suppose we assume that we need to test the significant differences between similar rainy months within the examined year. In that case, it cannot be implemented with the total monthly rainfall since each month contains one value.

(3) It seems that the two research sites have different landcover. Since soil moisture is also affected by plants, this factor should be properly addressed.

Response: both sites are covered with native desert plants. The natural site is covered with native desert plants where Rhanterietum epapposum is the dominant species. However, the irrigated site is also revegetated with native desert plants, which also include Rhanterietum epapposum. Therefore, they have the same effect on soil moisture. We also rephrased the study area to make it clearer. See line 84-87.

(4) Only one year (2017) measure data is available, the results can be questioned. This part should be explained.

Response: thank you for this great comment. We agree that it was not clearly discussed and needs a better explanation. The actual data were taken in the year 2017, before and after the rainfall season. They were used to validate the performance of the model in the year 2017 only. Also, the data used in the model, such as rainfall and evaporation, are actual data. However, the only predicted variable was evapotranspiration and runoff. The model used to estimate the runoff was also implemented in similar areas and showed promising results in previous studies. We add a few sentences in the methodology section to explain the validation process to make it clearer. We also revised the caption in figure 4 and the results. Please see line 118-121 and line 279.

(5) Both model-predicted and measured soil moisture data were used in this study, in the results and discussion, which data was referer to should be clearly stated. For example, in the regression analysis, measured or predicted soil moisture data was used? L195: it seems that the modeled results were used in the regression analysis. If this was the case, the results can be.

Response: yes, the predicted soil moisture was used in the regression analysis since the actual data was used to validate the model's performance for the year 2017.

Reviewer 2 Report

This paper addresses the effectiveness of irrigation to improve soil moisture in an arid region, and models out into the future based upon a max and min range of predicted climate patterns. I found the paper interesting and pertinent to concerns about desertification, water use, and water scarcity. My comments are mostly grammatical/formatting in nature.

My one main topical question refers to the thesis of the paper that soil moisture is not affected by summer irrigation. I understand that the authors are observing the effects of irrigation on soil moisture, and have determined that summer irrigation does not result in additional soil moisture. But have the authors considered the biomass output of the vegetation in the summer with/without irrigation? Does summer irrigation prevent the dessication/death of vegetation? Is the cost of maintaining vegetation worth the irrigation? just some questions to ponder. I admit that i am not familiar with this region, so this may be a moot point.

Line 43: "As a result of the anthropogenic activities..." I would suggest re-wording to "result of these anthropogenic activities"

Line 81: change "agriculture" to "agricultural"

Lines 81-82: make summer and winter plural "very hot summers and cold winters with limited..."

Line 83: Suggest adding "(perennial shrub)" after Rhanterietum epapposum, as this is the first reference to this name, and remove the "perennial shrub" after the second reference on line 87.

Line 85: suggest rewording "it has sand, loamy sand.." to "using drip irrigation in a Sand, Loamy Sand Sand soil."

Line 93: Figure 1: forgive me if I am mistaken, but i believe it should be "Saudi Arabia" on the map, instead of "Saudi Arabian." But that is from my western viewpoint, if it is referred to differently in the local region, then I see no reason to change it.

Line 98: reword to "for the ten-years period from...."

Line 118: Is there a citation for the JMP software that you can add to the references? It might be also good to list which version of the software was used.

Line 150: change "desalination" to "desalinated" 

line 151: i suggest changing L/day/m2 to "L/m2/day" so that you are basically saying "volume, per area, per temporal increment"

line 196: "TM" should be in superscript like in line 117

Line 216: make "year" plural. "based on the baseline years 1986-2005."

Line 257: make "year" plural. "average rainfall data from the years 2008-2017"

Lines 258/259: i suggest describing the years in order. For the above average years, you are listing 2017 then 2015, and for the average years, you are listing 2016 before 2014. I recommend writing it to show 2015/2017 and 2014/2016.

Line 341: replace "continuance" with "continued"

Line 467: REFERENCES: Please add DOIs where available. A few references include them, but most do not, and they are recent enough to have them.

Thank you for your submission, and best of luck!

Author Response

Reviewer 2:

This paper addresses the effectiveness of irrigation to improve soil moisture in an arid region, and models out into the future based upon a max and min range of predicted climate patterns. I found the paper interesting and pertinent to concerns about desertification, water use, and water scarcity. My comments are mostly grammatical/formatting in nature.

My one main topical question refers to the thesis of the paper that soil moisture is not affected by summer irrigation. I understand that the authors are observing the effects of irrigation on soil moisture, and have determined that summer irrigation does not result in additional soil moisture. But have the authors considered the biomass output of the vegetation in the summer with/without irrigation? Does summer irrigation prevent the dessication/death of vegetation? Is the cost of maintaining vegetation worth the irrigation? just some questions to ponder. I admit that i am not familiar with this region, so this may be a moot point.

Response: Thank you for your comment. All revegetated plants are considered perennial plants, which are mostly desert shrubs. The perennial shrubs don't die during the off-season (hot seasons). They turn into dry woody plants during the off-season where temperature increases and rainfall decreases, and flourish again during the spring season. Therefore, for sure, summer irrigation will not prevent vegetation's desiccation/death, and the cost of maintaining vegetation by irrigating the desert is not worth it. The findings of this work are essential since several restoration and revegetation projects are planning to consider irrigation as an option to support native desert plants' growth. We believe that irrigation needs to be considered carefully, especially in desert ecosystems, with limited water resources.

Line 43: "As a result of the anthropogenic activities..." I would suggest re-wording to "result of these anthropogenic activities"

Response: Revised as suggested. See line 43.

Line 81: change "agriculture" to "agricultural"

Response: Revised as suggested. Please see line 82.

Lines 81-82: make summer and winter plural "very hot summers and cold winters with limited..."

Response: Revised as suggested. Please see line 83-84.

Line 83: Suggest adding "(perennial shrub)" after Rhanterietum epapposum, as this is the first reference to this name, and remove the "perennial shrub" after the second reference on line 87.

Response: Revised as suggested. Please see line 84.

Line 85: suggest rewording "it has sand, loamy sand.." to "using drip irrigation in a Sand, Loamy Sand Sand soil."

Response: Revised as suggested. Please see line 86.

Line 93: Figure 1: forgive me if I am mistaken, but i believe it should be "Saudi Arabia" on the map, instead of "Saudi Arabian." But that is from my western viewpoint, if it is referred to differently in the local region, then I see no reason to change it.

Response: Revised as suggested. Please see Figure 1.

Line 98: reword to "for the ten-years period from...."

Response: Revised as suggested. Please see line 99.

Line 118: Is there a citation for the JMP software that you can add to the references? It might be also good to list which version of the software was used.

Response: Revised as suggested. Please see line 119.

Line 150: change "desalination" to "desalinated" 

Response: Revised as suggested. Please see line 151.

line 151: i suggest changing L/day/m2 to "L/m2/day" so that you are basically saying "volume, per area, per temporal increment"

Response: Revised as suggested. Please see line 152.

line 196: "TM" should be in superscript like in line 117

Response: Revised as suggested. Please see line 197.

Line 216: make "year" plural. "based on the baseline years 1986-2005."

Response: Revised as suggested. Please see line 217.

Line 257: make "year" plural. "average rainfall data from the years 2008-2017"

Response: Revised as suggested. Please see line 259.

Lines 258/259: i suggest describing the years in order. For the above average years, you are listing 2017 then 2015, and for the average years, you are listing 2016 before 2014. I recommend writing it to show 2015/2017 and 2014/2016.

Response: Revised as suggested. Please see line 259-260.

Line 341: replace "continuance" with "continued"

Response: Revised as suggested. Please see line 341.

Line 467: REFERENCES: Please add DOIs where available. A few references include them, but most do not, and they are recent enough to have them.

Response: Revised as suggested.

Thank you for your submission, and best of luck!

Reviewer 3 Report

This is a well-crafted article. We just lack the practicality of this research and its applicability. In a practical setting, authors should also consider other modelling options. I lack a more detailed specification of the surface runoff. If water retention measures are introduced in the area, this will limit surface runoff, how would this work in the model? Have you tried the balance equation from other authors, which also includes the relief of the territory? You have tried modelling in a GIS environment, there are very good tools for analyzing the area and for assessing the surface runoff, what changes will occur by changing the layout of the area and the construction of water retention measures. At present, very few models are dealing with this and climate change is already happening, we should just find out how to adapt to it and retain as much rainwater as possible in our territory. Evaporation is unavoidable but surface runoff should be kept to a minimum. It would be appropriate to add another scenario in order to improve surface runoff conditions and improve water retention in the area.

Author Response

Reviewer 3:

This is a well-crafted article. We just lack the practicality of this research and its applicability. In a practical setting, authors should also consider other modelling options. I lack a more detailed specification of the surface runoff. If water retention measures are introduced in the area, this will limit surface runoff, how would this work in the model? Have you tried the balance equation from other authors, which also includes the relief of the territory? You have tried modelling in a GIS environment, there are very good tools for analyzing the area and for assessing the surface runoff, what changes will occur by changing the layout of the area and the construction of water retention measures. At present, very few models are dealing with this and climate change is already happening, we should just find out how to adapt to it and retain as much rainwater as possible in our territory. Evaporation is unavoidable but surface runoff should be kept to a minimum. It would be appropriate to add another scenario in order to improve surface runoff conditions and improve water retention in the area.

Response: Thank you for this excellent and exciting comment. I agree that GIS has ideal tools that can be implemented to analyze and assess surface runoff, which is applicable to large scale studies covering different soils and topographical features. However, our study was implemented in small and flat areas with a low amount of rainfall, leading to shallow runoff events. The study areas are also flat areas and mostly covered with the same soil type, which will not show any significant runoff variations. Thus, these tools could be considered in our future work to assess the runoff significantly to improve surface runoff and water retention in areas with high runoff events.v

Round 2

Reviewer 1 Report

The manuscript presents a study on the influence of irrigation on the soil moisture status in the arid ecosystem. The authors thoroughly improved the manuscript in this revised version and most of the responses are convincing and clarifying. In general, the resubmitted manuscript is better structured and well-written. I think this manuscript is ready to be moved forward.

Author Response

Reviewer 1

The manuscript presents a study on the influence of irrigation on the soil moisture status in the arid ecosystem. The authors thoroughly improved the manuscript in this revised version and most of the responses are convincing and clarifying. In general, the resubmitted manuscript is better structured and well-written. I think this manuscript is ready to be moved forward.

Response: We would like to thank the reviewer for his effort in helping to improve the paper.